# Activated polymorphonuclear derived extracellular vesicles are potential biomarkers of periprosthetic joint infection

Imre Sallai [1]*, Nikolett Marton[2], Attila Szatmári[1], Ágnes Kittel[3], György Nagy[4,5], Edit I. Buzás[5,6,7], Delaram Khamari[5], Zsolt Komlósi[5], Katalin Kristóf[8], László Drahos[9], Lilla Turiák[9], Simon Sugár[9], Dániel Sándor Veres[10], Daniel Kendoff[11], Ákos Zahár[12], Gábor Skaliczki[1]

1 Department of Orthopaedics, Semmelweis University, Budapest, Hungary, 2 Jahn Ferenc Hospital, Budapest, Hungary, 3 Institute of Experimental Medicine, Budapest, Hungary, 4 Department of Rheumatology and Clinical Immunology, Semmelweis University, Budapest, Hungary, 5 Department of Genetics, Cell- and Immunobiology, Semmelweis University, Budapest, Hungary, 6 HCEMM Extracellular Vesicle Research Group, Semmelweis University, Budapest, Hungary, 7 ELKH-SE Immune-Proteogenomics Extracellular Vesicle Research Group, Budapest, Hungary, 8 Clinical Microbiological Diagnostic Laboratory, Semmelweis University, Budapest, Hungary, 9 MS Proteomics Research Group, Research Centre for Natural Sciences, Budapest, Hungary, 10 Department of Biophysics and Radiation Biology, Semmelweis University, Budapest, Hungary, 11 Helios Klinikum Berlin-Buch, Berlin, Germany, 12 Helios Klinikum Emil von Behring, Berlin, Germany

* sallai.imre@semmelweis-univ.hu

**Data Availability Statement:** All relevant data are within the paper and its Supporting information files.

## Abstract

### Background

Extracellular vesicles (EVs) are considered as crucial players in a wide variety of biological processes. Although their importance in joint diseases or infections has been shown by numerous studies, much less is known about their function in periprosthetic joint infection (PJI). Our aim was to investigate activated polymorphonuclear (PMN)-derived synovial EVs in patients with PJI.

### Questions/Purposes

(1) Is there a difference in the number and size of extracellular vesicles between periprosthetic joint aspirates of patients with PJI and aseptic loosening? (2) Are these vesicles morphologically different in the two groups? (3) Are there activated PMN-derived EVs in septic samples evaluated by flow cytometry after CD177 labelling? (4) Is there a difference in the protein composition carried by septic and aseptic vesicles?

### Methods

Thirty-four patients (n = 34) were enrolled into our investigation, 17 with PJI and 17 with aseptic prosthesis loosening. Periprosthetic joint fluid was aspirated and EVs were separated. Samples were analysed by nanoparticle tracking analysis (NTA) and transmission electron microscopy (TEM) and flow cytometry (after Annexin V and CD177 labelling). The protein content of the EVs was studied by mass spectrometry (MS).

**Funding:** This work was supported by the ENDO-Verein Hamburg, Germany. (and EFOP-3.6.3-VEKOP-16-2017-00009). The funders had no role in study design, data collection and analysis, decision to publish, or preparation of the manuscript.

**Competing interests:** The authors have declared that no competing interests exist.

**Abbreviations:** ABP, annexin binding puffer; ACD-A, acid citrate dextrose A; AHSG, alpha-2-HS-glycoprotein; EV, extracellular vesicle; FC, flow cytometry; IC, immune complex; MIC, minimum inhibitory concentration; MS, mass spectrometry; NTA, nanoparticle tracking analysis; PBS, phosphate-buffered saline; PJI, prosthetic joint infection; PMN, polymorphonuclear neutrophils; SF, synovial fluid; TEM, transmission electron microscopy; THA, total hip arthroplasty; TKA, total knee arthroplasty.

## Results

NTA showed particle size distribution in both groups between 150 nm and 450 nm. The concentration of EVs was significantly higher in the septic samples (p = 0.0105) and showed a different size pattern as compared to the aseptic ones. The vesicular nature of the particles was confirmed by TEM and differential detergent lysis. In the septic group, FC analysis showed a significantly increased event number both after single and double labelling with fluorochrome conjugated Annexin V (p = 0.046) and Annexin V and anti-CD177 (p = 0.0105), respectively. MS detected a significant difference in the abundance of lactotransferrin (p = 0.00646), myeloperoxidase (p = 0.01061), lysozyme C (p = 0.04687), annexin A6 (p = 0.03921) and alpha-2-HS-glycoprotein (p = 0.03146) between the studied groups.

## Conclusions

An increased number of activated PMN derived EVs were detected in the synovial fluid of PJI patients with a characteristic size distribution and a specific protein composition. The activated PMNs-derived extracellular vesicles can be potential biomarkers of PJI.

## Introduction

In the past few years, the importance of extracellular vesicles (EVs) in cell-to-cell communication was widely investigated. Numerous physiological and pathological processes were identified, in which EVs play an important role. These include i) inflammatory responses (e.g. COPD—inflammatory airway obstruction and emphysema) [1, 2], ii) blood clotting processes [3], iii) tissue repair [4], iv) pregnancy [5], v) autoimmune diseases [6], vi) cardiovascular diseases [7], vii) hematological diseases [8], viii) cancer [9] or ix) infection [10]. EVs isolated from the synovial fluid have been shown to have a unique pattern, specific to different joint diseases [11, 12].

One of the most challenging complications in orthopedic surgery is the periprosthetic joint infection (PJI), often necessitating multiple surgeries, longer rehabilitation and accompanied by major economic consequences. Therefore, distinction between septic and aseptic implant failures is of paramount importance. To date, there is no single examination for the diagnosis of PJI, instead, a set of different tests are used [13]. Very few data are available about the role of EVs in PJI. The aim of this research was to detect EVs in the periprosthetic synovial fluid (SF) and investigate the possibility to use them as new biomarkers, which may provide a novel diagnostic approach and may contribute to a better understanding of PJI. Further aim of our study was detect, observe and compare the membrane properties, the transported cargo and the morphology of the detected EVs from previously proven septic and aseptic samples were also performed.

Our main questions were the follows: Is there a difference in the number and size of extracellular vesicles between periprosthetic joint aspirates of patients with PJI and aseptic loosening? Are these vesicles morphologically different in the two groups? Are there activated PMN-derived EVs in septic samples evaluated by flow cytometry after CD177 (activated human PMNs marker) labelling? Is there a difference in the protein composition carried by septic and aseptic vesicles?

## Patients and methods

Our single-center prospective comparative study was conducted in the Orthopedic Department of the Semmelweis University (Budapest, Hungary). The study was approved by the Semmelweis University Regional and Institutional Committee of Science and Research Ethics (Ethics approval number: SE TUKEB number: 4/2015); all patients signed an informed consent before enrollment. The study was performed in accordance with the declaration of Helsinki, and the MISEV 2018 guidelines were followed [14]. Due to the different volumes of the obtained synovial fluid samples, not all samples could be analyzed by all methods.

### Patients

A total of 34 patients were enrolled into the investigation. The study cohort consisted of 17 patients with PJI, that was further divided to two groups. Eight patients (7 females, 1 male; 4 total hip arthroplasties (THAs), 4 total knee arthroplasties (TKAs); average age: 75.3±9.4 years) with an early onset infection (onset of the infection < 90 days from the surgery) [15] formed the acute septic group, and nine patients (4 females, 5 males; 6 THAs, 3 TKAs; average age: 73.1±11.4 years) with low-grade or chronic infection (onset of the infection > 90 days from the surgery) [15] created the low-grade septic group. Patients did not receive antibiotic treatment before the sampling. Seventeen patients scheduled for surgery with aseptic loosening (15 females, 2 males; 13 THAs, 4 TKAs; average age: 70.8±8.3 years) formed the aseptic control group. Aseptic status based on in all cases previous and intraoperative negative cultures and laboratory data. PJI was determined by the current definition of the Musculoskeletal Infection Society and International Consensus Meeting on Periprosthetic Joint Infection criteria [13] (Table 1).

### Isolation of Extracellular Vesicles (EVs)

Synovial fluid (SF) was obtained during revision surgery: after exposure of the joint and before capsulotomy, SF was aspirated by a needle (21 gauge) through the intact capsule. The sample was divided in two: a minimum of 3 mL was used for EV isolation, and the rest of the sample was sent to microbiological examination in hemoculture bottles (BACT/ALERT® Culture Media, bioMerieux, Durham, USA).

Immediately after aspiration, SF was injected into an ACD-A laboratory sample collecting tube, a tube that prevents *in vitro* release of EVs by cells [16]. The sample was centrifuged for 15 minutes (using BOECO Centrifuge C-28A 2500 RCF) to sediment macroscopic elements and the supernatant was removed with a thin (18 gauge) needle for further analysis. The highly

**Table 1. Demographic and laboratory data of the involved patients.**

| | septic | | aseptic |
|---|---|---|---|
| | acute | low grade | |
| patients (n = 34) | 8 | 9 | 17 |
| | (7 women; 1 men) | (4 women; 5 men) | (15 women; 2 men) |
| age (years; mean ± SD) | 75.3 ± 9.4 | 73.1 ± 11.4 | 70.8 ± 8.3 |
| affected joint (hip/knee) | 4 hip / 4 knee | 6 hip / 3 knee | 13 hip / 4 knee |
| CRP (mg/L; mean) | 156 | 61 | 6.7 |
| We (mm/h; mean) | 71 | 54.3 | 17.3 |
| WBC (G/L; mean) | 13.1 | 10.2 | 5.9 |
| Neutrofil % | 79.1 | 64.6 | 57.8 |
| MSIS/ICM criteria score | two positive cultures (major criteria) | two positive cultures (major criteria) | all ≤ 3 |

viscous SF was digested by 1500 U/mL hyaluronidase (MERCK, Hyaluronidase from sheep testes type II, lyophilized powder, ≥300 units/mg) for 30 minutes at room temperature. Afterwards, the sample was diluted two folds by PBS, (pre-filtered by a 0.2 μm syringe filter MERCK Puradisc Syringe filter), and was centrifuged at 2,500g for 8 minutes. Filtration was then performed by two different sized syringe filters, first, a 1.2 micrometer-sized, then a 0.8 micrometer-sized filter (MERCK Puradisc Syringe filter). Special care was taken to use solely gravity driven filtration since any extra pressure could have damaged our target vesicles. EVs were pelleted from the filtered samples at 12.500g for 60 minutes at room-temperature. The supernatant was removed with a thin needle (18 gauge), the pellet was resuspended for flow cytometry (FC) in 150 μL annexin binding puffer (ABP) and pre-filtered in a 0.2 μm syringe filter. The prepared samples were stored at -30 ˚C until use.

## Nanoparticle tracking analysis (NTA)

Nanoparticle tracking analysis was performed in three (n = 3) septic and four (n = 4) aseptic samples. To reach an appropriate particle concentration range, NTA samples were diluted in 1 mL of PBS. The analysis of particle size distribution and concentration was performed on Zeta-View PMX120 instrument (Particle Metrix, Germany). For all measurements, 11 cell positions were scanned at 25˚C (in 2 cycles) with the following camera settings: shutter speed-100, sensitivity-75, frame rate-7.5, video quality—medium (30 frames). The analysis of the videos was performed by the ZetaView Analyze software 8.05.10 with a minimum area of 5, maximum area of 1000, and a minimum brightness of 20.

## Transmission electron microscopy (TEM)

Six samples (2 aseptic, 2 low-grade infection and 2 acute purulent) were collected and analyzed by TEM. The aim of the TEM analysis was to visualize the vesicles in all types of samples (sterile, low-grade infection, acute infection) and investigate their properties.

The EV-containing pellet was fixed with 4% paraformaldehyde in PBS (pH 7. 5) for 2 hours at room temperature, rinsed with PBS and postfixed in 1% $OsO_4$ for 15 min. After a short rinse with distilled water, the pellet was dehydrated in graded ethanol including block-staining with 1% uranyl acetate in 50% ethanol for 15 min, finally was embedded in Taab 812 (Aldermaston, T031 UK). Following overnight polymerization at 60˚C (Taab 812), 60 nm ultrathin sections were cut by a Leica UCT ultramicrotome (Leica Microsystems, UK) and analyzed with a Hitachi7100 (Hitachi, Japan) electron microscope equipped by Veleta, a 2 k × 2 k Mega-Pixel side-mounted TEM CCD camera (Olympus). Contrast and brightness of electron micrographs were adjusted by Adobe Photoshop CS3 (Adobe Photoshop Incorporation, CA).

## Flow cytometry (FC) analysis of samples

FC was applied to detect activated human polymorphonuclear (PMN) cell-derived EVs. Ten aseptic (n = 10), five (n = 5) acute septic and three (n = 3) low-grade septic samples were stained with 2 μL FITC-conjugated Annexin V (Sony Biotechnology, San Jose, USA)–extracellular vesicle marker—and 2 μL PE-conjugated anti-human CD177 (n = 10) (Sony Biotechnology, San Jose, USA)—activated human PMNs marker. All samples were incubated for 40 minutes at 4˚C in a dark incubator, then diluted with 350 μL ABP, and were incubated for further 20 minutes at 4˚C in a dark incubator. For FC detection, a FACSCalibur flow cytometer (Becton Dickinson) was used with the following settings: flow rate was held under 1000 events/s; FSC = E01 (log); SSC = 890V (log); Fl-1 = 381V (log). For calibration of flow-cytometry $1x10^6$ beads/ml 1.0 μm reference calibration beads were used (Invitrogen, ThermoFisher Scientific). Visualization of FC data was performed by Kaluza Analysis software Version 2.1.

### Differential detergent lysis

In order to differentiate EVs from non-vesicular particles, all samples were treated by Triton X-100 detergent (0.1% for 60 seconds) after the first flow cytometry measurement [17]. The difference between the event numbers before and after detergent lysis provided the real EV event number, so this value was used for further analysis throughout the study.

### Mass spectrometry (MS)

Mass spectrometry was performed to explore the cargo of the EVs in 9 aseptic, 6 acute septic and 7 low-grade septic samples. Proteins were extracted from the EV samples by repeated freeze-thaw cycles [18]. Ten μg aliquots were digested using an in solution tryptic digestion protocol [19]. Five hundred ng digests were analyzed on a Dionex Ultimate 3000 nanoRSLC (Dionex, Sunnyvale, Ca, USA) coupled to a Bruker Maxis II mass spectrometer (Bruker Daltonics GmbH, Bremen, Germany) via CaptiveSpray nano booster ion source [20]. Raw data files were processed using the Compass DataAnalysis software (Bruker, Bremen, Germany). Protein identification and label-free quantitation were performed using MaxQuant software (version 1.6.17.0) [21] using its Andromeda search engine. During the Andromeda search, a focused database was used. The focused database was created following protein search on Byonic software (v3.6.0, Protein Metrics Inc, San Carlos, CA, USA) against the Swissprot Homo sapiens database applying the following search criteria: precursor mass tolerance 20 ppm, fragment mass tolerance 40 ppm, cleavage at lysine and arginine C terminal, maximum 2 missed cleavages, and 1% FDR limit. Modifications were set as follows: Carbamidomethyl/+57.021464 @ C | fixed; Oxidation/+15.994915 @ M | common1; Glu->pyro-Glu/-18.010565 @ NTerm E | common1.

### Statistical analysis

Statistical analysis and graphic illustrations were conducted by the Perseus statistical software and GraphPad software. Differences among study groups were analyzed after calculating normal distribution by T-test, one-way ANOVA and post hoc Tukey statistical tests. Difference was considered significant if p-value was <0.05, results of the FC and MS analysis are shown in mean±SD format.

## Results

### Bacterial culture of the samples

Microbiological examination of joint fluid aspirates from the septic group (acute and low-grade) yielded the following bacteria: methicillin sensitive *Staphylococcus aureus* (MSSA) in 4 cases, methicillin sensitive *Staphylococcus epidermidis* (MSSE) in 2 cases, *Staphylococcus hominis* in 2 cases and 1–1 cases of *Escherichia coli*, *Streptococcus pneumoniae*, *Streptococcus dysgalactiae*, vancomycin resistant *Enterococcus faecalis* (VRE), *Cutibacterium acnes*, *Massilia variant*, *Staphylococcus haemolyticus*, *Staphylococcus warneri*, *Streptococcus agalactiae* (Fig 1). Identical bacteria grew from the samples harvested and submitted to culture preoperatively.

### The septic samples containing larger size and higher concentration of EVs proved by nanoparticle tracking analysis (NTA)

After the analysis of 4 aseptic and 3 septic samples, the number of EVs were different between the two groups; there was a significantly higher concentration in the septic group as compared

## Bacteria isolated from septic samples

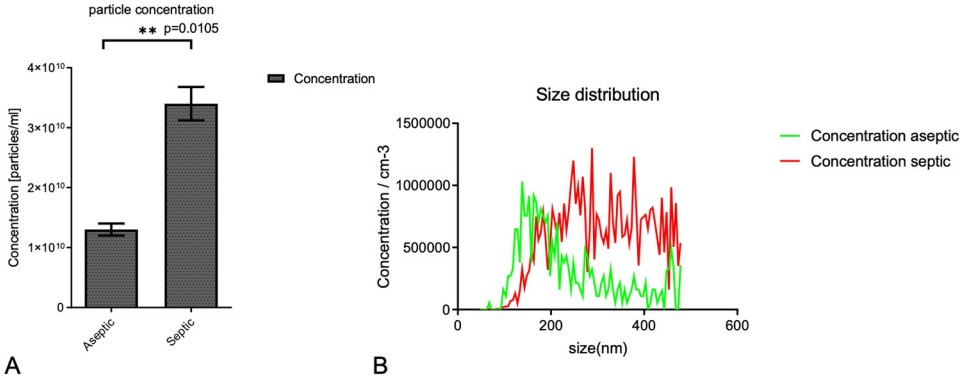

**Fig 1. Bacteria yielded from the synovial fluid of patient with PJI.** (Orange–low-grade infection, red–acute infection).

to the aseptic group (p = 0.0105, T-test) (Fig 2A). Besides the differences in concentration of EVs, the size distribution was also different; particle size ranges in both groups were between 150–450 nm. The septic group showed a shift towards the larger sizes and formed multiple peaks between 300 nm and 400 nm range. The aseptic samples had a different size distribution pattern with the highest concentration between 150–200 nm and a decreased number of particles in the larger size range. (Fig 2B).

**Fig 2. NTA analysis showed a significantly higher number of EVs in the septic group than in aseptic samples.** (p = 0.0105, T-test) (A). The different size distribution of EVs analysed by NTA. Green line represents aseptic samples, red line refers to septic aspirates (B).

## The EVs from septic samples have different morphological properties under TEM

After standard preparation and fixation, 2 aseptic, 2 low-grade septic and 2 acute septic samples were investigated under TEM. In all groups we found vesicular structures. However, we observed substantial numerical and morphological differences. When compared to the septic aspirates, less vesicles were found in aseptic samples with a size range between 150–200 nm. EVs in this group had a thin, regular membrane and only a few contained some dense cargo inside. The highest concentration of EVs was found in the acute septic samples, showing vesicles in the size range of 350–400 nm, surrounded by a thick, often irregularly shaped membrane and containing a dense material inside. The same dense substance could also be observed around the EVs. Low-grade septic samples provided a mixture of acute and sterile aspirates showing EVs between 150–300 nm in a concentration between aseptic and acute septic aspirates. The vesicles were less regularly shaped than in the aseptic samples and the previously described dense material could be investigated mostly around the vesicles rather than inside of them. (Fig 3A–3C).

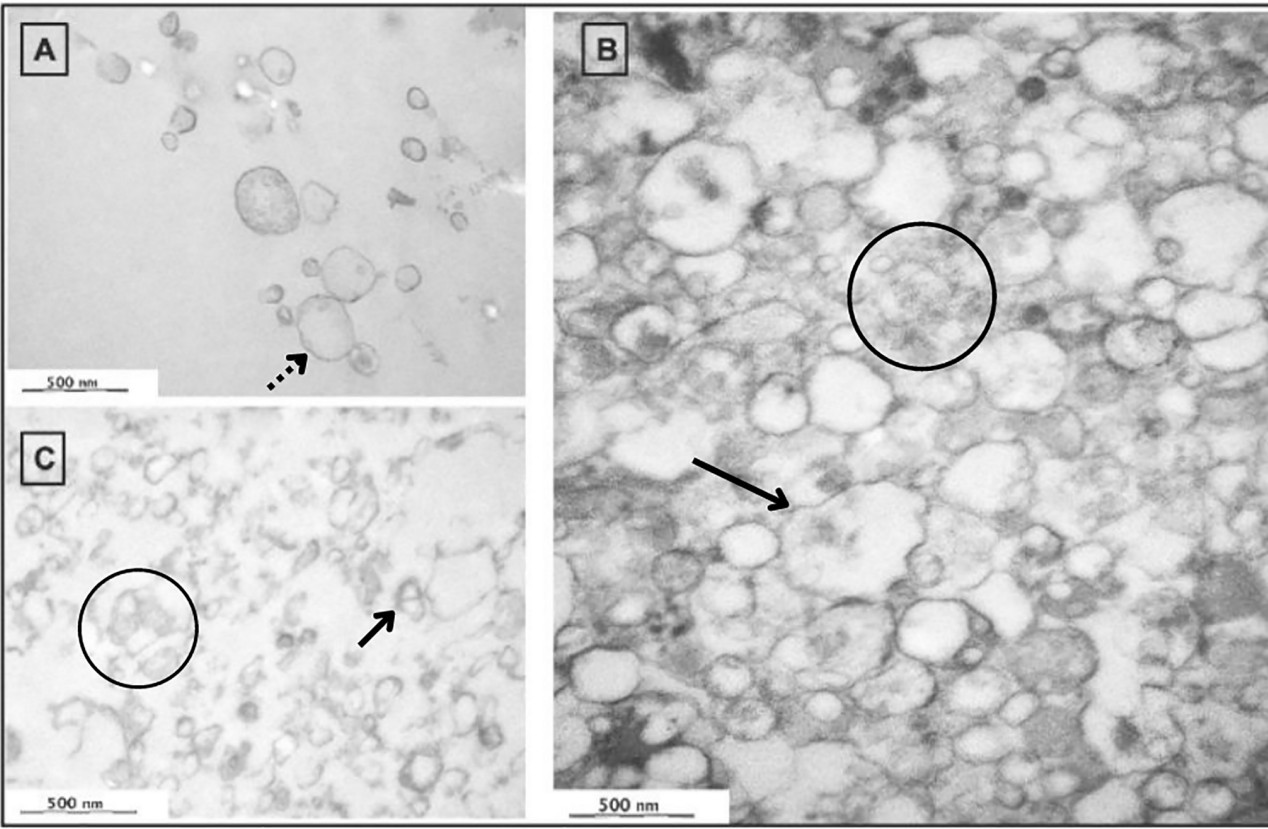

**Fig 3.** Smaller vesicles in a lower concentration, surrounded by a thin membrane (dashed line arrow) could be observed during the TEM analysis of aseptic samples (A). Acute septic samples (B) showed a markedly higher number of vesicles with a thin, irregularly shaped membrane (arrow) and containing a dense cargo (circle). The same dense material could be seen also outside the vesicles (circle). Low-grade septic samples (C) provided more vesicles than aseptic aspirates, the EVs were larger than aseptic particles with a thick wall (arrow) and a dense substance inside them (circle). (Magnitude: 30x and the markers show the 500 nm size range).

### The CD177 expressed by the activated PMNs appeared in greater amounts on the surface of vesicles from the septic sample

Ten aseptic and 10 septic (6 low-grade and 4 acute) samples were investigated. After staining with FITC-conjugated Annexin V, our FC analysis resulted in 2480 ± 1096 events in the septic group and 1521 ± 1308 events in the aseptic group, the difference was significant (p = 0.046, T-test). (Fig 4A and 4B). There was no significant difference between the acute and low-grade septic samples (p = 0.0799 and p = 0.0815, T-test) (Fig 4C and 4D). Double labelling with FITC-conjugated Annexin V and PE-conjugated anti-human CD177 showed significantly higher number of events in the septic group than that of the aseptic group (1753 ± 648 vs. 648 ± 541 events, respectively, p = 0.000809, T-test). All measured events were in the 200–800 nm size range. The event number decreased and fell down to the range of instrument noise range in both groups after detergent lysis. (Fig 5).

### Further potential biomarkers found inside the EVs by Mass Spectrometry (MS) analysis

Nine aseptic, 6 acute septic and 7 low-grade septic samples were investigated by MS. The cumulated EV protein concentration was 6.65 ± 6.02 μg/μl in the acute septic group, 5.3 ± 4.67 μg/μl in the low-grade septic group and 4.3 ± 3.32 μg/μl in the aseptic group. The differences between the groups were not significant. (Fig 6).

Three hundred and eighty proteins were identified in the three groups. After normalization of the concentrations, those proteins were selected for further statistical analysis (ANOVA and FDR correction) which were presented in at least two-third of the samples. (Fig 7).

Of the investigated proteins, alpha-2-HS-glycoprotein was detected in a significantly higher concentration in the aseptic samples than in the septic aspirates, while the concentration of lactotransferrin, myeloperoxidase, lysozyme C and annexin A6 was significantly higher in the septic group than that of the aseptic group. (Figs 8 and 9).

## Discussion

The information about the presence and role of EVs in different biological processes is continuously growing and their implication in infections are also widely investigated [1, 22–26]. A better understanding of their function or their use as biomarkers could open new therapeutic opportunities and may possibly serve as a novel approach in the differential diagnosis in pathological conditions such as aseptic and septic implant failures.

In our study, three different patient cohorts were investigated: acute septic, low-grade or chronic septic and aseptic groups. To make the differential diagnosis between these groups as clear as possible, the current recommendations of the Musculoskeletal Infection Society and International Consensus Meeting on Periprosthetic Joint Infection were used [13]. Challenges of working with biological samples include the differences in the amount of sample and sample quality. In addition, separation of EVs has a few pitfalls that can impede the correct analysis. Therefore, a compliance with consensus protocols is essential. To this end, the MISEV 2018 guidelines were followed throughout our work [14]. Because of the unequal amount of SF harvested from different patients, not all samples could be analyzed by all methods.

A very few data are available about the role of PMNs derived EVs in prosthetic joint infections. A significant increase in the release of PMN-derived EVs has been shown after bacterial opsonization by by Tímár et al [26]. Furthermore, Rüwald et al also reported similar data investigating SF of PJI patients [27]. The same results were provided by our study: our NTA analysis revealed a significantly higher particle concentration in the septic samples than in

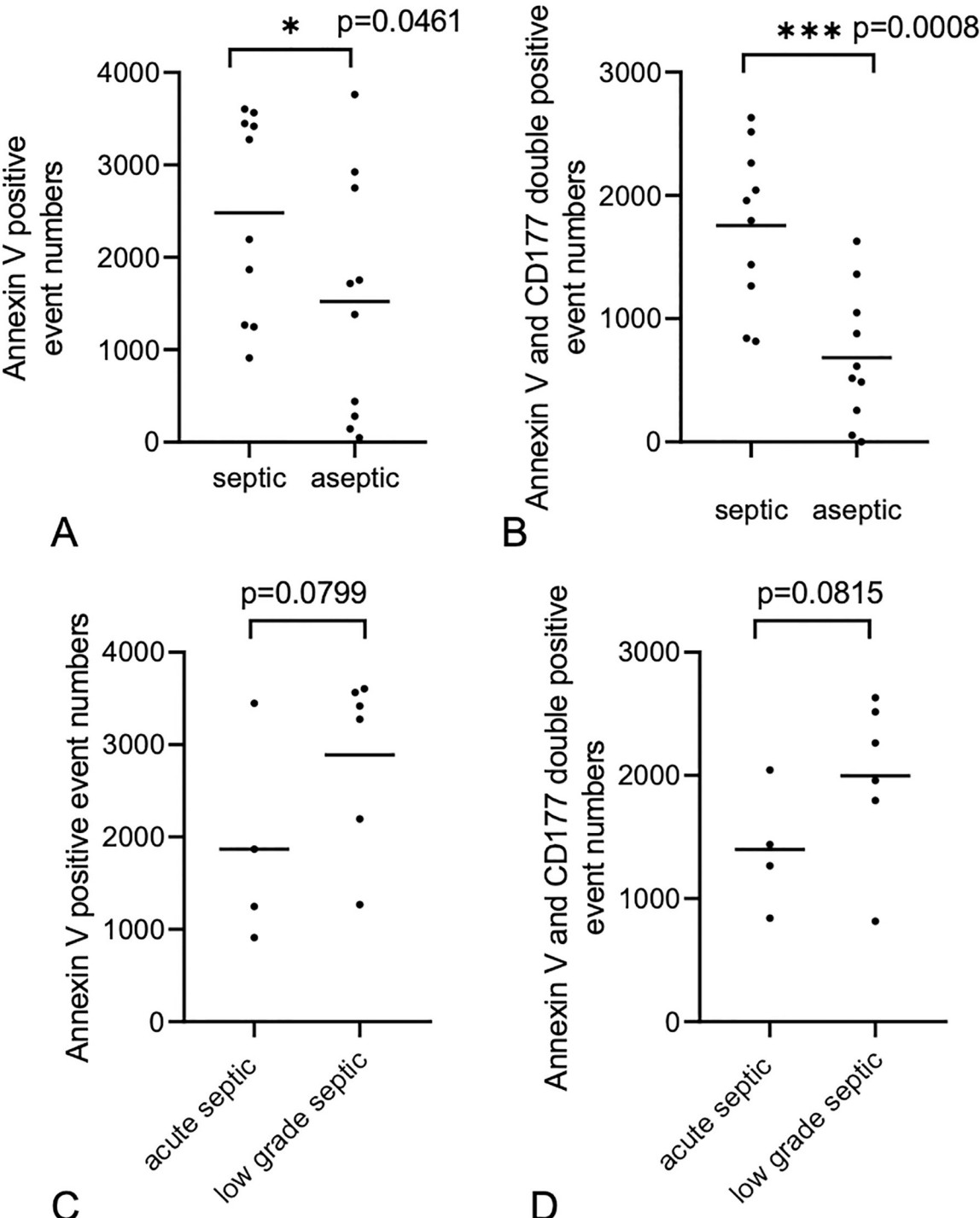

**Fig 4. The Annexin V conjugated EVs event numbers were significantly higher in the septic group (p = 0.046, T-test) than is the aseptic group.** The Annexin V and CD177 double stained event number was also significantly higher (p = 0.000809, T-test) in the septic than in the aseptic group (A-B). There was no significant difference between the acute and low-grade septic samples (C-D).

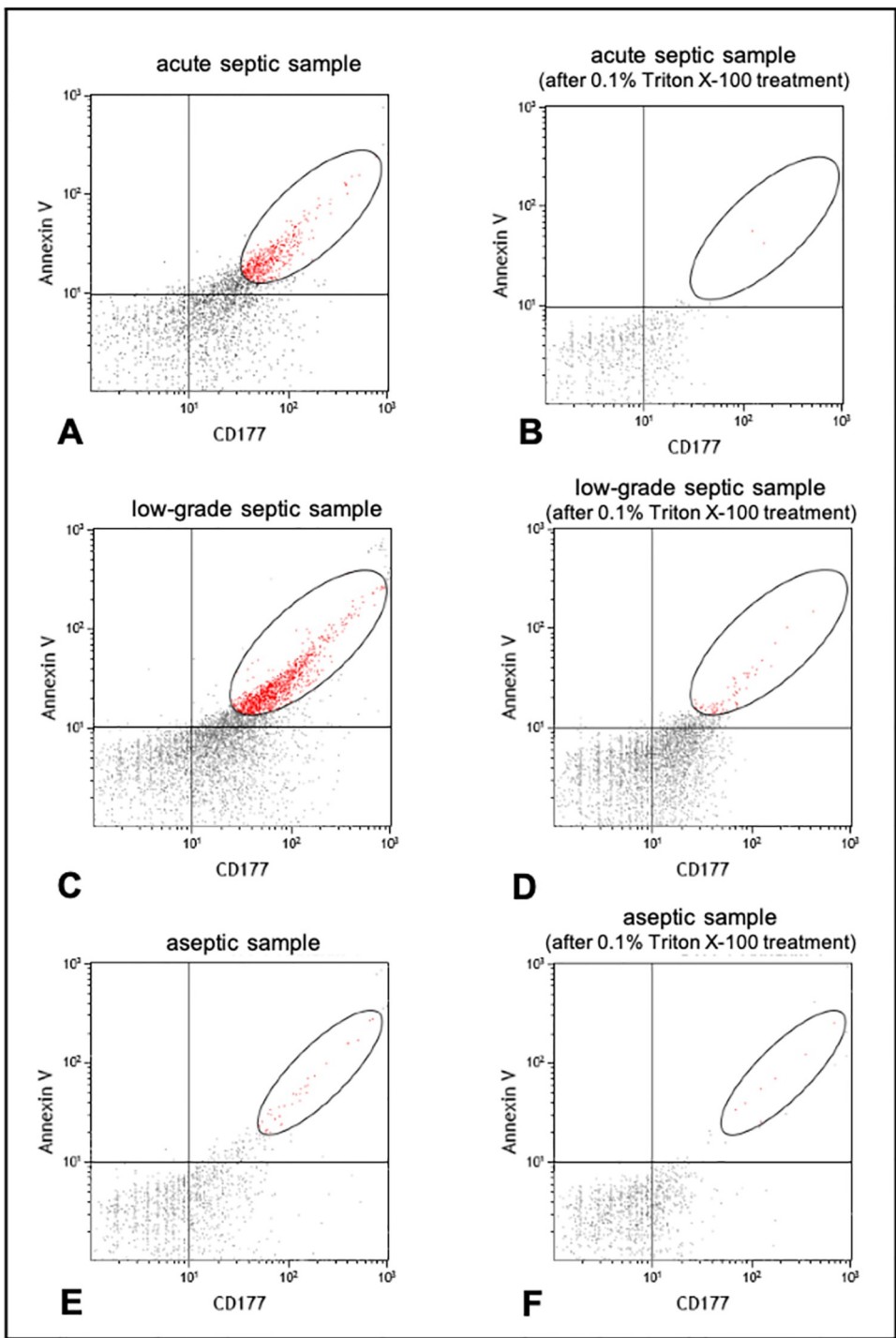

**Fig 5.** Dot plots depict the FC analysis of acute septic (A) low-grade septic (C) and aseptic (E) samples labelled with annexin V and CD177 surface markers. After 0.01% Triton X-100 detergent lysis (B,D,E) real vesicular events disappear. The difference in the FC event numbers before and after detergent lysis represents the true number of EVs. The oval gates show the size range of the EVs.

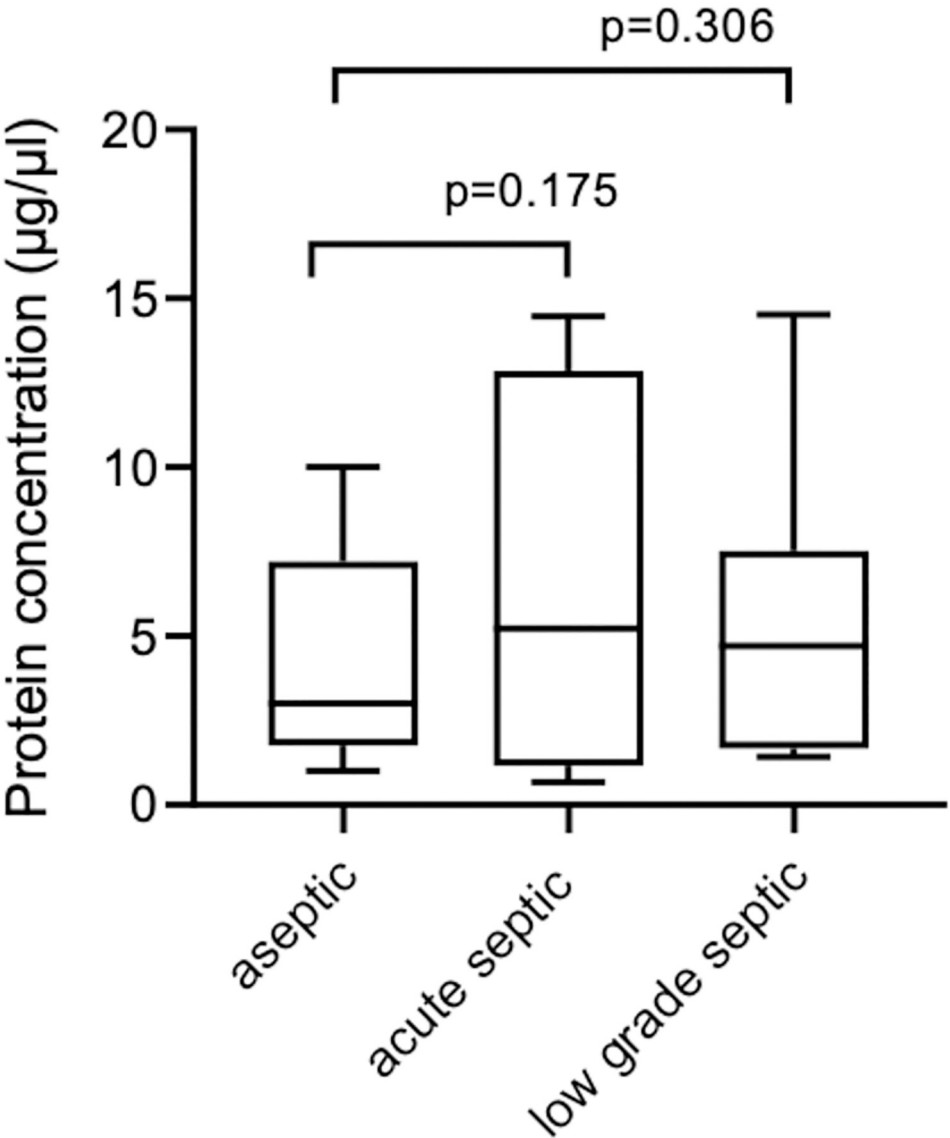

**Fig 6. The cumulated EV protein concentration (ug/ul) in the acute septic group was the highest, followed by the low-grade septic group and the aseptic group.** The differences were not significant (p = 0.175 and p = 0.306).

aseptic aspirates. The size distribution also differed between the two groups: EVs in aseptic samples peaked in a lower, 150–200 nm range while EVs from septic aspirates had the highest concentration between 300–450 nm. This is in contrast with the findings of Rüwald et al. since they observed smaller particle sizes in septic samples. However, they used a different EV isolation protocol (ultracentrifugation 100,000x g for 2 hours), which could explain the differences.

Our TEM analysis showed morphological differences of EVs in the three examined groups. Sterile EVs could be described by a regularly shaped, thin membrane, while EVs from septic samples had irregularly shaped, thick membrane and these vesicles often contained a dense cargo material.

In the SF of PJI patients, presumably vesicles of different origin can be detected, such those released by bacteria, granulocytes, macrophages, stem cells, etc. Thus, staining of EVs before

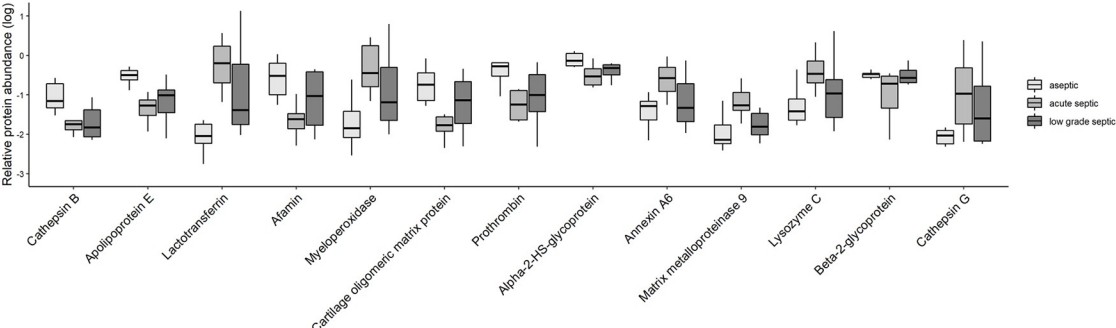

**Fig 7. The relative EV protein abundance (log) in the three groups.** Boxes represent the relative abundance of certain identified proteins after logaritmization. Dots represent the individual values and their distribution; the mean value is shown inside the box (black line).

FC analysis is essential for determining the cells that release them. Annexin V binding is a common EV feature. In our study, saline with suitable $Ca^{2+}$ content was used for ideal Annexin V staining [28]. As PJI is an infectious process, PMN-derived EVs were in the center of our interest. CD177 is a surface marker expressed exclusively by neutrophils in 40–60% of healthy donors, its upregulation was observed in different inflammatory and infectious diseases and also in patients with bacteremia suggesting its role in the activation of PMNs by infection [26, 29–31]. This is why we used immunostaining in our study with anti-CD177 antibodies as well.

When interpreting the results of FC analysis, it is important to confirm the vesicular nature of the measured events, since any particle with the similar biophysical properties (size, light scattering, sedimentation) would appear during flow cytometry. As it was previously reported by György et al., insoluble ICs overlap with EVs resulting in overestimation of the number of vesicles [28]. Moreover, antibodies and antibody aggregates can also form protein complexes and interfere the results [22]. An easy but still reliable method to discriminate EVs in the size range of 100-1000n (medium sized EVs) from protein-complex related events is to use detergents at a concentration that is sufficient to lyse EVs but does not dissemble ICs and protein aggregates [22, 32]. Hence, in our study Triton X-100 detergent lysis was used. With this method, the difference in the FC event numbers before and after detergent lysis represented the true number of EVs.

In our study ten septic and ten sterile samples were investigated. Both groups were labeled with the same fluorescent proteins (Annexin V and anti-CD177) using the same protocol. Both single staining with Annexin V and double labelling with Annexin V and CD177 resulted a significantly increased number of EVs in the septic group as compared to the aseptic samples (p = 0.046, p = 0.000809, respectively, T-test).

Previous investigations analyzing proteins in the joint aspirate of PJI patients generally used the supernatant of the harvested sample. Shahi et al published a list of proteins as possible new biomarkers for the diagnosis of PJI including neutrophil elastase, bactericidal/permeability-increasing protein and neutrophil gelatinase-associated lipocalin [33]. The authors emphasized the potentially powerful but yet not fully predictable role of biomarkers in the diagnosis of implant associated infections. Deirmengian et al. suggested human α-defensin 1–3, neutrophil elastase 2, bactericidal/permeability-increasing protein, neutrophil gelatinase-associated lipocalin, and lactoferrin to be used as biomarkers for the diagnosis of PJI with 100% sensitivity and specificity [34]. Wang et al. screened proteins by MS in the SF of PJI patients, and found lactoferrin, polymorphonuclear leukocyte serine protease 3, and myeloid nuclear

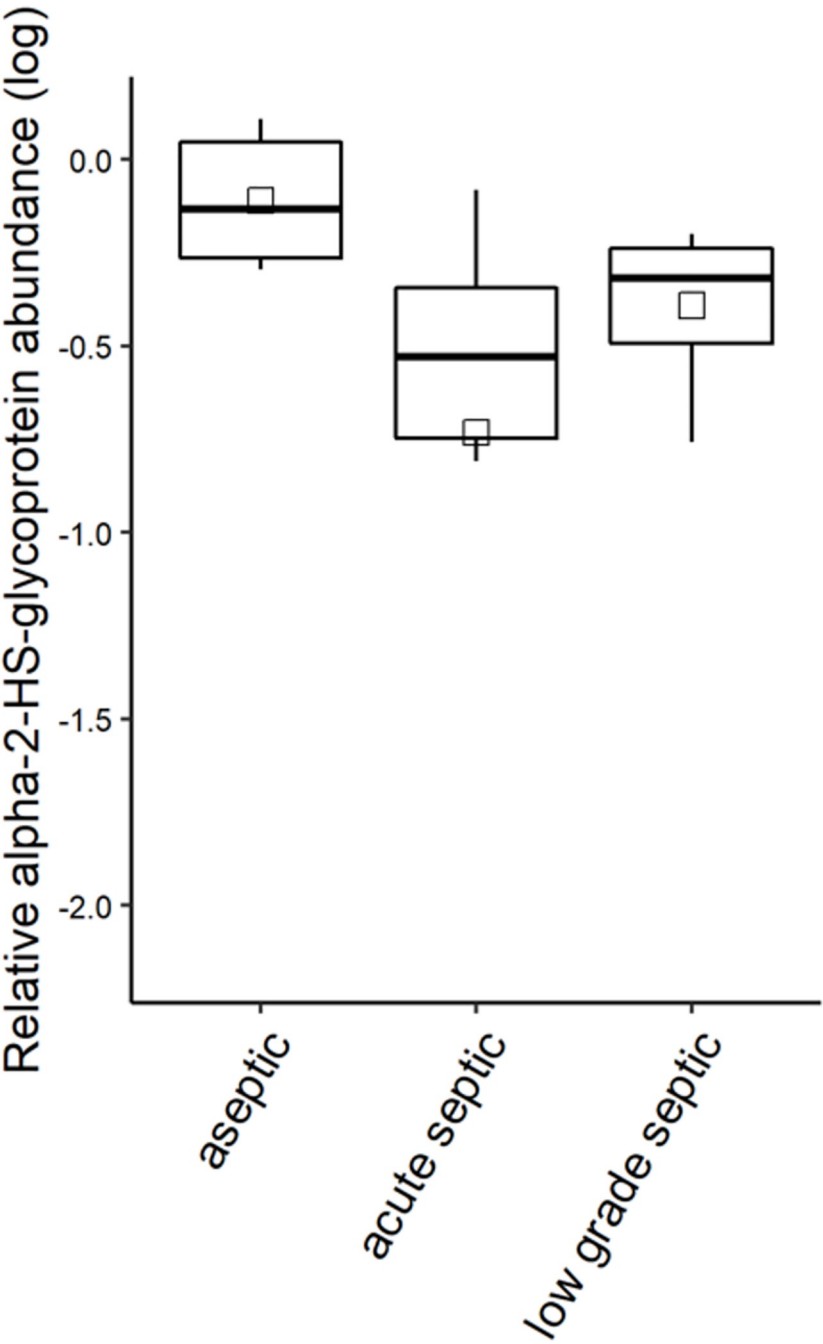

**Fig 8. The relative alpha-2-HS-glycoprotein abundance (log).** AHSG was presented in a significantly higher concentration in the sterile group, than in the acute septic or low-grade septic group. (p = 0.0314627, ANOVA test).

differentiation antigen as promising synovial biomarkers for differentiating between patients with and without PJI [35].

By contrast, in our study, intravesicular proteins extracted from EVs by repeated freeze-thaw cycles were investigated by MS. The following proteins were detected in a significantly

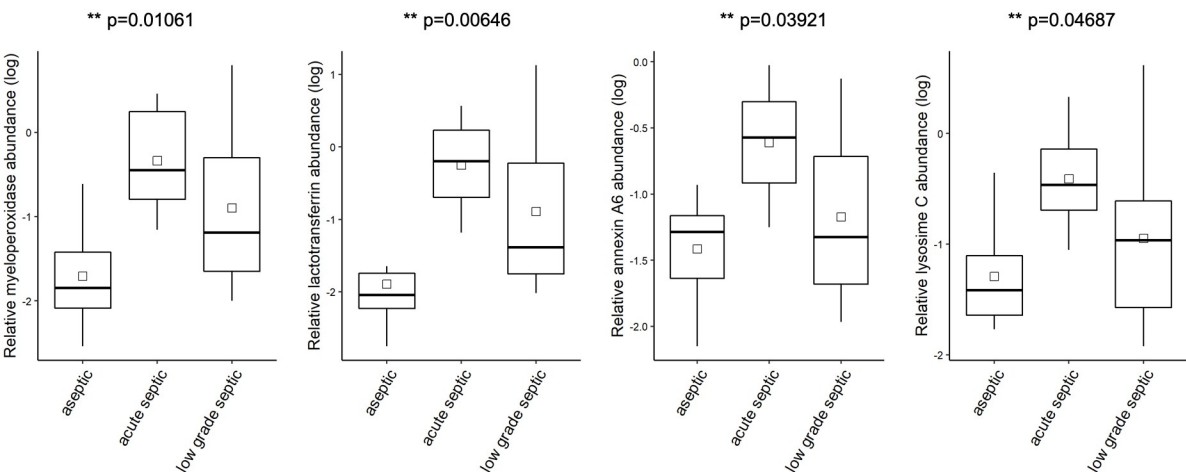

**Fig 9. The relative abundances (log) of the myeloperoxidase, lactotransferrin, annexin A6 and lysozyme C.** All proteins were found in a significantly higher concentration in the acute and low-grade septic group than in the sterile group. (myeloperoxidase p = 0.01061, lactotransferrin p = 0.00646, annexin A6 p = 0.03921, lysozyme C p = 0.04687) ANOVA test.

elevated concentration in septic samples as compared to aseptic aspirates: lactotransferrin (p = 0.00646), myeloperoxidase (p = 0.01061), lysozyme C (p = 0.04687) and annexin A6 (p = 0.03921) with one way ANOVA test.

We found one protein with an opposite change: alpha-2-HS-glycoprotein (AHSG) level was significantly higher (p = 0.03146 one-way ANOVA test) in the sterile group than in the acute septic or in the low-grade septic group. AHSG is a known negative acute phase protein with serum levels correlating inversely with serum C-reactive protein concentration [36, 37], which explains our finding.

In summary, our investigation showed that there is a significantly higher number of activated PMN-derived EVs in the SF of patients with PJI than in that of patients with aseptic implant loosening. The size distribution of these EVs differed also demonstrating a shift towards a larger size range in septic patients. TEM analysis showed morphological variations, as septic EVs were less regularly shaped, had a thicker membrane and contained a dense cargo. Larger particles with different morphological properties were observed in the septic group however these alterations were not statistically analysed, or proved to be significant. When investigating the protein content of the EVs, a significantly higher concentrations of lactotransferrin, myeloperoxidase, lysozyme C and annexin A6 were found in septic samples, while the level of AHSG was significantly higher in aseptic aspirates. Further investigations are needed to clarify if these results may open a novel approach in the diagnostics of PJI. Microfluidics and lab on chip techniques can be solutions to simplify and speed up our procedure and give an opportunity to create a point of care test. Detection of activated PMNs derived EVs from synovial fluid can be an alternative solution in culture negative cases. Detection of these activated PMNs derived EVs from blood serum could be the next generation of PJI diagnostics. However, this study led to a lot questions that deserved further investigation.

## Limitations

Our study has some limitations. First, the number of patients is relatively low, which could bias our results and underpower the statistical analysis. Furthermore, due to the different volume of the harvested synovial fluid, not all samples could be analysed by all methods.

However, basic research studies generally investigate small study groups [27] due to the nature of these intensive investigations. To strengthen our results in the future, we need to repeat our study on a larger cohort and define cut off values. Septic samples were obtained from patients with PJI caused by different microorganisms. Although the type of bacteria can influence the intensity of inflammation [33], the pathophysiology of the immune response is similar, supporting the applicability of our method in all PJI cases [38].

## Supporting information

**S1 Data.**
(XLS)

## Acknowledgments

We would like acknowledge the help of the Department of Laboratory Medicine (Semmelweis University, Budapest, Hungary) especially Barna Vásárhelyi, Ibolya Kocsis and Balázs Szalay.

On behalf of Project "Proteomikai BigData értékelése" we thank for the usage of ELKH Cloud (https://science-cloud.hu/) that significantly helped us achieving the results published in this paper.

## Author Contributions

**Conceptualization:** Imre Sallai, György Nagy, Edit I. Buzás, Zsolt Komlósi, Lilla Turiák, Gábor Skaliczki.

**Data curation:** Imre Sallai.

**Formal analysis:** Imre Sallai.

**Funding acquisition:** Gábor Skaliczki.

**Investigation:** Imre Sallai, Nikolett Marton, Attila Szatmári, Ágnes Kittel, Delaram Khamari, Katalin Kristóf, László Drahos, Lilla Turiák, Dániel Sándor Veres, Gábor Skaliczki.

**Methodology:** Imre Sallai, Nikolett Marton, Attila Szatmári, György Nagy, Gábor Skaliczki.

**Project administration:** Imre Sallai, Attila Szatmári.

**Software:** Imre Sallai, László Drahos, Lilla Turiák, Dániel Sándor Veres.

**Supervision:** György Nagy, Edit I. Buzás, Lilla Turiák, Daniel Kendoff, Ákos Zahár, Gábor Skaliczki.

**Validation:** Gábor Skaliczki.

**Visualization:** Imre Sallai, Delaram Khamari, László Drahos, Lilla Turiák, Simon Sugár, Dániel Sándor Veres.

**Writing – original draft:** Imre Sallai, Nikolett Marton, Attila Szatmári, Ágnes Kittel, György Nagy, Edit I. Buzás, Delaram Khamari, Zsolt Komlósi, László Drahos, Lilla Turiák, Simon Sugár, Dániel Sándor Veres, Gábor Skaliczki.

**Writing – review & editing:** Imre Sallai, Nikolett Marton, Attila Szatmári, Ágnes Kittel, György Nagy, Edit I. Buzás, László Drahos, Lilla Turiák, Daniel Kendoff, Ákos Zahár, Gábor Skaliczki.

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
