## [Decision Letter · Decision Letter 0]

14 Mar 2022

PONE-D-21-26411Activated polymorphonuclear derived extracellular vesicles are potential biomarkers of periprosthetic joint infectionPLOS ONE

Dear Dr. Sallai,

Thank you for submitting your manuscript to PLOS ONE. After careful consideration, we feel that it has merit but does not fully meet PLOS ONE’s publication criteria as it currently stands. Therefore, we invite you to submit a revised version of the manuscript that addresses the points raised during the review process.

We look forward to receiving your revised manuscript.

Kind regards,

Afsheen Raza, PhD

Academic Editor

PLOS ONE

Journal Requirements:

3. Thank you for stating the following financial disclosure: "This work was supported by the ENDO-Verein Hamburg, Germany. (and EFOP-3.6.3-VEKOP-16-2017-00009)"

5. Your abstract cannot contain citations. Please only include citations in the body text of the manuscript, and ensure that they remain in ascending numerical order on first mention.

6. Please upload a new copy of Figure 5 as the detail is not clear. Please follow the link for more information: https://blogs.plos.org/plos/2019/06/looking-good-tips-for-creating-your-plos-figures-graphics/" https://blogs.plos.org/plos/2019/06/looking-good-tips-for-creating-your-plos-figures-graphics/

7. Please include your tables as part of your main manuscript and remove the individual files. Please note that supplementary tables (should remain/ be uploaded) as separate "supporting information" files

Reviewers' comments:

Reviewer's Responses to Questions

**Comments to the Author**

1. Is the manuscript technically sound, and do the data support the conclusions?

Reviewer #1: Yes

Reviewer #2: Yes

2. Has the statistical analysis been performed appropriately and rigorously? 

Reviewer #1: Yes

Reviewer #2: Yes

3. Have the authors made all data underlying the findings in their manuscript fully available?

Reviewer #1: Yes

Reviewer #2: Yes

4. Is the manuscript presented in an intelligible fashion and written in standard English?

Reviewer #1: No

Reviewer #2: Yes

5. Review Comments to the Author

Reviewer #1: This is an interesting observational study showing elevated PMN derived EVs in in septic as compared to aseptic prosthetic joint aspirates. Such septic EVS are enriched for a number of PMN proteins such as MPO. Overall, the study is well done and the results are convincing.

Minor points:

1) In the Introduction when discussing EVs, the authors should cite a paper that is related , Specifically, they should cite KR Genschmer el al. Cell 176, 113-126, 2019. This paper describes exosome-sized EVs from activated PMNs that are found in COPD and likely cause disease.

2) In Figure 2 there are numerous peaks between 250 and 450 nm and they should be described as such. It seems erroneous to conclude that there are only peaks at 300 and 400 nm.

3) On page 13 lines 256 and 261, Figure 2A and B are reversed from the order found in the Figure legend.

4) As presented Figure 3 is not particularly helpful. It would be improved by adding arrows to indicate what is being described in the text such as "dense cargo", "thin regular membrane", etc.

5) There are numerous spelling errors that should be corrected.

Reviewer #2: In this study Sallai et al investigated extracellular vesicles (EV) from synovial fluid of patients with prosthetic joint inflammation and aseptic loosening. Their findings are valuable for researchers in this field and possibly for clinicians as well on the long run.

Although the methods used are of high scientific value and quality, but the relatively low number of patients enrolled and compared is a weakness.

The introduction section is clear and well written, although “entailing serious impact … on health care providers”, “analysing and comparison” could be better phrased.

Reference 8 is a review about miRNA in lung cancer, please replace it with a study that supports the role of EV in cancer.

The aims of the study are clearly stated.

The methods used are adequate.

The study cohort consisted of 17 patients, who were divided into two subgroups, which seem quite low. The methods used in this basic research study could justify this, and this weakness was addressed in the limitations section. How does this number compare to the studies cited in the introduction (Ref 1-11)?

The authors state, that not all samples have been analysed by all methods, due low volume of obtained synovial fluid. How did the authors decide which sample would be analysed by a certain method? How were patient numbers in these groups decided, they seem rather random.

Did the authors considered using less investigative methods, which would have allowed them to analyse more samples in those one or two methods?

Patient in the acute septic group never received antibiotics, they were immediately scheduled for revision surgery?

Table 1 shows patient groups, and some inflammation markers. Only these markers were used to assign patients to groups? Or the scoring system described in Ref 12, mentioned in line 147 was used? If so, then what were the exact scores for these groups?

I recommend not using empty rows for dividing the table, and in the fifth cell of the last column there is an “x” instead of “4”.

Throughout the study patients were divided into three groups and their data was analysed that way. However, for Nanoparticle tracking analysis they were divided into two groups? What is the explanation for this?

Larger particles with different morphological properties were observed in the septic group, however these alterations were not statistically analysed, or proved to be significant. This should be noted in the Discussion section.

The finding of larger particles in the septic group contradicts Ref 26, as noted by the authors. Any possible explanation for these conflicting results?

Has there been any study investigating CD177 expressing cells in PJI, to which the study results could be compared?

The summary is clean and concise, however discussing clinical application seems a bit far reaching to me.

The Funding statement is not clear for me. Data in parenthesis in the details of the ENDO-Verein support, or a separate funding?

Other remarks

Alpha-2-HS-glycoprotein is usually abbreviated as AHSG, and not AHSGP.

Typos, grammatical errors: line 116, 142, 322, 324, 406, 408

6. PLOS authors have the option to publish the peer review history of their article (what does this mean?). If published, this will include your full peer review and any attached files.

Reviewer #1: No

Reviewer #2: No

---

## [Author Response · Author response to Decision Letter 0]

19 Apr 2022

All responses are written in the „response to reviewers” file attached in.

---

## [Editor Report · Decision Letter 1]

22 Apr 2022

Activated polymorphonuclear derived extracellular vesicles are potential biomarkers of periprosthetic joint infection

PONE-D-21-26411R1

Dear Dr. Sallai,

We’re pleased to inform you that your manuscript has been judged scientifically suitable for publication and will be formally accepted for publication once it meets all outstanding technical requirements.

Kind regards,

Afsheen Raza, PhD

Academic Editor

PLOS ONE
---

## [Editor Report · Acceptance letter]

29 Apr 2022

PONE-D-21-26411R1 

Activated polymorphonuclear derived extracellular vesicles are potential biomarkers of periprosthetic joint infection. 

Dear Dr. Sallai:

I'm pleased to inform you that your manuscript has been deemed suitable for publication in PLOS ONE. Congratulations! Your manuscript is now with our production department. 

Kind regards, 

on behalf of

Dr. Afsheen Raza 

Academic Editor

PLOS ONE